# The first North American *Propterodon* (Hyaenodonta: Hyaenodontidae), a new species from the late Uintan of Utah

Shawn P. Zack

Department of Basic Medical Sciences, University of Arizona College of Medicine –Phoenix, Phoenix, AZ, United States of America

School of Human Evolution and Social Change, Arizona State University, Tempe, AZ, United States of America

## ABSTRACT

The carnivorous mammalian fauna from the Uintan (late middle Eocene) of North America remains relatively poorly documented. This is unfortunate, as this is a critical interval in the transition from "creodont" to carnivoran dominated carnivore guilds. This study reports a new species from the Uinta Formation of the Uinta Basin, Utah, the first North American species of the otherwise Asian hyaenodont genus *Propterodon*. The new species, *Propterodon witteri*, represented by a dentary with $M_{2-3}$ from the late Uintan Leota Quarry, is larger than the well-known *P. morrisi* and *P. tongi* and has a larger $M_3$ talonid, but is otherwise very similar. A phylogenetic analysis of hyaenodont interrelationships recovers *P. witteri* as a hyaenodontine but is generally poorly resolved. A relationship between Hyaenodontinae and *Oxyaenoides*, recovered by many recent analyses, is not supported. Among the Asian species of *Propterodon*, *P. pishigouensis* is reidentified as a machaeroidine oxyaenid and recombined as *Apataelurus pishigouensis* new combination. *Isphanatherium ferganensis* may also represent an Asian machaeroidine. Identification of a North American species of *Propterodon* and an Asian *Apataelurus* increases the similarity of North American Uintan and Asian Irdinmanhan faunas and suggests that there was substantial exchange of carnivorous fauna during the late middle Eocene.

## INTRODUCTION

Hyaenodonts are a significant component of Eocene carnivorous guilds across the Holarctic and Africa (*Gunnell, 1998*; *Rose, 2006*; *Lewis & Morlo, 2010*). Along with other "creodonts" (e.g., Oxyaenidae), hyaenodonts are distinguished from modern carnivorans and their fossil relatives (Carnivoraformes) by the presence of multiple carnassial pairs in the dentition, which results in alternating shearing and crushing/grinding areas in the dentition, rather than regional separation of the molar series into mesial shearing and distal crushing/grinding areas. The latter innovation in Carnivoraformes, and convergently in Viverravidae (*Zack, 2019*), may have facilitated the ecological diversification of carnivorans (*Friscia & Van Valkenburgh, 2010*) ultimately allowing carnivorans to displace hyaenodonts

Corresponding author
Shawn P. Zack,
zack@email.arizona.edu

over the course of the Paleogene in the northern continents and Miocene in Africa (*Wesley-Hunt, 2005*; *Friscia & Van Valkenburgh, 2010*; *Borths & Stevens, 2017*).

In North America, hyaenodont diversity was greatest during the earlier half of the Eocene, particularly the Wasatchian and Bridgerian North American Land Mammal Ages (NALMAs) (*Gunnell, 1998*; *Van Valkenburgh, 1999*; *Wesley-Hunt, 2005*; *Friscia & Van Valkenburgh, 2010*). In the subsequent Uintan NALMA, hyaenodont diversity declined dramatically. Only four genera, *Limnocyon*, *Mimocyon*, *Oxyaenodon*, and *Sinopa*, have been described from Uintan faunas (*Matthew, 1899*; *Matthew, 1909*; *Peterson, 1919*; *Gustafson, 1986*), although an additional, small hyaenodont taxon is known but undescribed (*Rasmussen et al., 1999*; S Zack, pers. obs., 2019). This mid-Eocene decline of hyaenodont and other "creodont" diversity corresponds with an increase in the diversity of carnivorans and their immediate relatives (Carnivoraformes) (*Van Valkenburgh, 1999*; *Wesley-Hunt, 2005*; *Friscia & Van Valkenburgh, 2010*), a pattern suggesting some form of replacement of hyaenodonts by carnivoraform taxa. Understanding the nature of that replacement requires a detailed record of the diversity of both groups.

Reexamination of existing collections is one key to refining the record of carnivorous mammals across this critical period, as overlooked or misidentified specimens can shift the temporal and geographic ranges of known taxa and allow recognition of new forms. MCZ VPM 19874, the specimen that forms the focus of the present study, is an example of significant discoveries that can be made in existing collections. The specimen, a dentary with $M_{2-3}$, was collected by a Harvard University expedition to the Uinta Basin, Utah in 1940 (Fig. 1) and has not been described or mentioned in the literature in almost 80 subsequent years. It documents a new hyaenodont taxon from the late Uintan that differs substantially from known Uintan hyaenodonts, particularly in its possession of a strongly hypercarnivorous morphology, greater than previously known in Wasatchian through Uintan North American hyaenodonts. In fact, the affinities of the new taxon appear to lie with *Propterodon*, a genus previously known only from eastern Asian faunas correlated with the Chinese middle Eocene Irdinmanhan and Sharamurunian stages (*sensu Wang et al., 2019*). The new taxon increases Uintan hyaenodont diversity and disparity while providing evidence for interchange of Asian and North American carnivores during this critical interval in the divergent histories of Hyaenodonta and Carnivoraformes.

## MATERIALS & METHODS

Dental terminology follows *Rana et al. (2015)*, with two exceptions. "Mesiobuccal cingulid" is used following *Zack (2011)* instead of "buccal cingulid", as this structure is mesially restricted in the new species. Following *Kay (1977)*, "hypocristid" is used rather than "postcristid" for the crest connecting the hypoconid and hypoconulid. Measurements follow *Gingerich & Deutsch* (*1989*, fig. 1) and *Borths & Seiffert* (*2017*, fig. 1e), with the addition of a measurement of maximum talonid height. Dental measurements taken are illustrated in Fig. 2. Mandibular depth was measured lingually below $M_3$. All measurements were taken to the nearest tenth of a millimeter with Neiko digital calipers. MCZ VPM 19874 was whitened using ammonium chloride prior to being photographed.

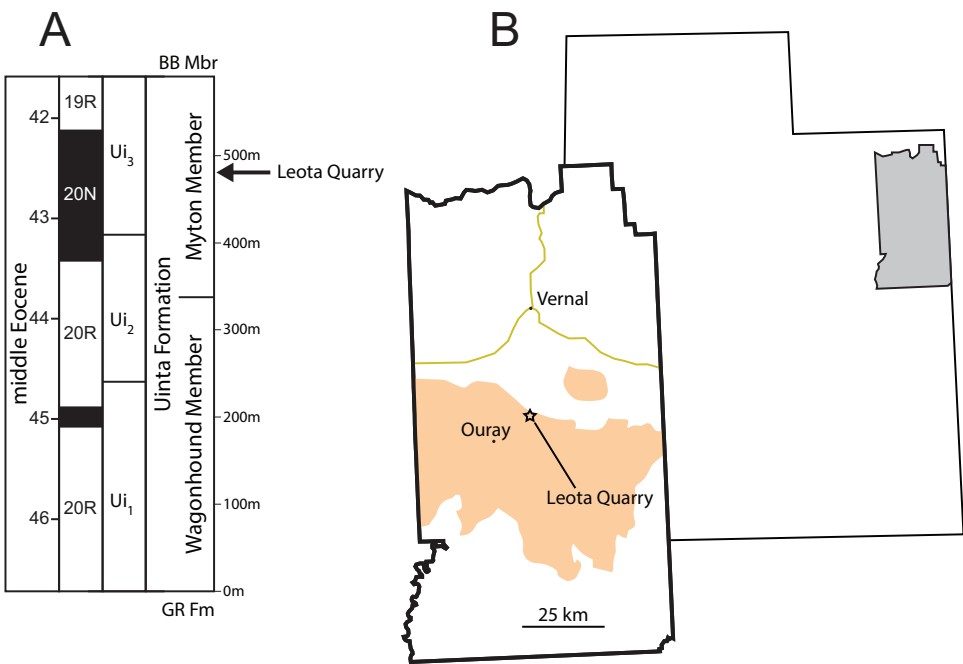

**Figure 1** **Stratigraphic and geographic position of Leota Quarry.** (A) Generalized stratigraphic section of middle Eocene Uinta Formation in the west-central Uinta Basin showing the position of Leota Quarry along with biochron boundaries (*Prothero, 1996*) and geomagnetic polarity chrons (*Murphey et al., 2018*). (B) Map of Utah, United States showing the location of Uintah County and map of Uintah County showing the position of Leota Quarry (as indicated by *Peterson & Kay, 1931*). Orange shading in (B) indicates outcrop of the Uinta Formation (after *Hintze, 1980*). Abbreviations: BB Mbr, Brennan Basin Member of the late middle Eocene Duchesne River Formation; Gr Fm, early middle Eocene Green River Formation. Drawings by Shawn P. Zack.

The electronic version of this article in Portable Document Format (PDF) will represent a published work according to the International Commission on Zoological Nomenclature (ICZN), and hence the new names contained in the electronic version are effectively published under that Code from the electronic edition alone. This published work and the nomenclatural acts it contains have been registered in ZooBank, the online registration system for the ICZN. The ZooBank LSIDs (Life Science Identifiers) can be resolved and the associated information viewed through any standard web browser by appending the LSID to the prefix http://zoobank.org/. The LSID for this publication is: urn:lsid:zoobank.org:pub:CDA777EE-C052-4922-90DD-AAFD41D3F345. The online version of this work is archived and available from the following digital repositories: PeerJ, PubMed Central and CLOCKSS.

**Phylogenetic Methods**—To test the taxonomic affinities of the new species, it was added to a substantially modified version of the character taxon matrix used by *Rana et al. (2015)*. The dental sample used by *Rana et al. (2015)* was modified to eliminate non-independent characters (e.g., removing a character describing the number of $P^3$ roots, which reflects development of a $P^3$ protocone lobe), following the recommendations of recent authors who have argued that inclusion of non-independent characters can mislead

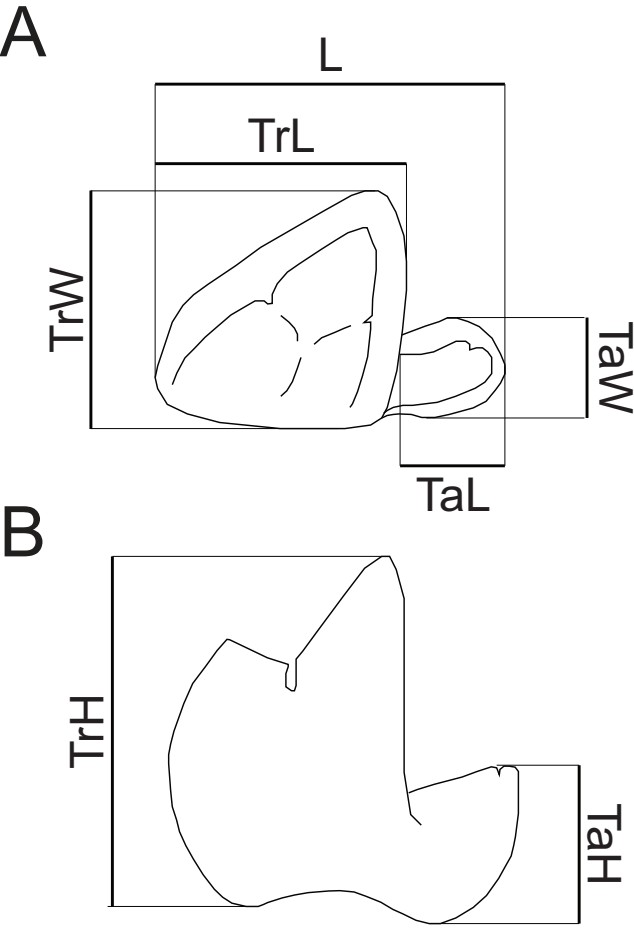

**Figure 2** **Measurements of hyaenodont lower molars.** Schematic drawing of a hyaenodont lower molar in (A) occlusal and (B) buccal views to show measurements taken for this study. Abbreviations: L, maximum length; TrL, maximum trigonid length; TrW, maximum trigonid width; TrH, maximum trigonid height; TaL, maximum talonid length; TaW, maximum talonid width; TaH, maximum talonid height. Drawings by Shawn P. Zack.

phylogenetic analyses that rely heavily on mammalian dental morphology (*Sansom, Wills & Williams, 2017*; *Billet & Bardin, 2019*). Overall, several dental characters were revised, replaced, combined, or deleted, and one additional character describing the number of upper incisors was added from *Borths & Stevens (2019a)*. Numerous individual scorings were modified to improve scoring consistency, with particular emphasis placed on ensuring scoring consistency across geographic regions.

While the dental character sample from *Rana et al. (2015)* was used, the non-dental character sample used by *Rana et al. (2015)* which, in turn was derived from *Polly (1996)*, was largely replaced by the cranial, mandibular, and postcranial character sample used by *Borths & Stevens (2019a)*, and Borths and Stevens' scorings were used with some additions (e.g., postcranial scorings were added for *Galecyon chronius* and *Prototomus martis*). One

character from *Rana et al. (2015)* describing mandibular symphysis depth was retained because this variation was not captured by Borths and Stevens' characters.

In addition to the inclusion of the new species, several changes were made to the taxonomic composition of the matrix. First, the composite *Propterodon* spp. OTU used by *Rana et al. (2015)* was replaced with separate OTUs for *P. morrisi* and *P. tongi*. Reflecting newly published material, the African "*Sinopa*" OTU included in *Rana et al. (2015)* was replaced by *Brychotherium ephalmos*, scored from descriptions in *Borths, Holroyd & Seiffert (2016)* and accompanying 3D models. Scorings of *Akhnatenavus* were updated to include *A. nefertiticyon* described in the same work, while scorings for *Masrasector* were updated based on material of *M. nananubis* described by *Borths & Seiffert (2017)*. The *Pterodon* spp. OTU was restricted to *P. dasyuroides* and rescored, given that new evidence indicates *Pterodon*, as traditionally defined, is likely polyphyletic (*Solé et al., 2015a*; *Borths & Stevens, 2019a*; *Borths & Stevens, 2019b*). Three additional taxa were added to the matrix, *Boritia duffaudi*, *Preregidens langebadrae*, and *Matthodon menui*. These three taxa are either newly described or newly identified as hyaenodonts, and they significantly enhance the documentation of early European hyaenodonts (*Solé, Falconnet & Yves, 2014*; *Solé, Falconnet & Vidalenc, 2015*).

In addition, six OTUs included in the *Rana et al. (2015)* matrix were excluded from the present analysis. As with *Pterodon*, monophyly of *Metapterodon*, as used by *Rana et al. (2015)*, now appears dubious (*Morales & Pickford, 2017*; *Borths & Stevens, 2019b*), but, unlike the well-documented *Pterodon dasyuroides*, individual species of *Metapterodon* are fragmentary and poorly known, contributing little to the broader structure of hyaenodont interrelationships. Until the composition of *Metapterodon* is better understood, the genus is better excluded. A second taxon, *Eoproviverra eisenmanni*, was removed over concerns about the permanent versus deciduous status of the type and most informative specimen, MNHN.F.RI 400. Described as an $M_2$ (*Godinot, 1981*; *Solé et al., 2015b*), MNHN.F.RI 400 shows several features that suggest the tooth may instead represent $dP_4$, including a low paraconid, open trigonid, small talonid, and generally tall, delicate cusp construction. If this is the case, MNHN.F.RI 400 would likely represent a larger taxon than the remainder of the hypodigm.

Finally, *Tinerhodon disputatum* and the three species that have been referred to Koholiinae (*Boualitomus marocanensis*, *Koholia atlasense*, *Lahimia selloumi*) were excluded. As briefly noted by *Rana et al. (2015)*, the hyaenodont status of these taxa remains to be clearly demonstrated. Referral of all four taxa to Hyaenodonta appears to have been made based on the presence of multiple carnassial pairs and retention of three molars. As discussed by *Zack (2019)*, this *de facto* definition of Hyaenodonta combines two eutherian symplesiomorphies (molar homodonty and three molars) with a trait found in all carnivorous clades (carnassials). Given this weak evidence, the possibility that some or all these taxa are not hyaenodonts must be considered. In fact, *Tinerhodon disputatum* has not been consistently recovered as a hyaenodont in analyses that do not constrain the ingroup to monophyly (e.g., *Borths & Stevens, 2019b*). Among members of the potentially polyphyletic Koholiinae, two species known exclusively from lower dentitions (*Boualitomus marocanensis* and *Lahimia selloumi*) lack $P_1$, a feature that is unusual for Hyaenodonta but

typical for members of Tenrecoidea (*Gheerbrant et al., 2006*; *Solé et al., 2009*). Combined with the small size of both species, this raises the possibility that koholiines may actually represent an endemic African carnivorous radiation prior to an Eocene immigration of hyaenodonts to Africa. The third koholiine, *Koholia atlasense*, is known only from a fragmentary upper dentition, and recent phylogenetic analyses have not recovered it in a clade with *B. marocanensis* and *L, selloumi* (*Borths, Holroyd & Seiffert, 2016*; *Borths & Seiffert, 2017*; *Borths & Stevens, 2017*; *Borths & Stevens, 2019a*; *Borths & Stevens, 2019b*). The $M^1$ of *K. atlasense* has a paracone that is distinctly lingual to the metacone, although this may be exaggerated by damage to the metacone (*Crochet, 1988*). This morphology is not characteristic of hyaenodonts but occurs in the early tenrecoids *Sperrgale minutus* and *Arenagale calcareus* (*Pickford, 2015*). Other aspects of the morphology of *K. atlasense* are also unusual for a hyaenodont including the elongate $P^4$ metastyle, strong $M^1$ prevallum shear, and massive $M^1$ parastyle connected to the preparacrista at its mesial margin. The overall morphology of *K. atlasense* is distinctive enough to cast doubt on its hyaenodont status.

The final matrix includes 48 ingroup taxa and two outgroups scored for 115 characters. The list of characters and specimens examined are available in the Supplemental Information. The full matrix is also available on MorphoBank as project P3489 (http://morphobank.org/permalink/?P3489). The matrix was analyzed using parsimony in TnT version 1.5 (*Goloboff & Catalano, 2016*). Initial analyses used the Sectorial Search algorithm under the New Technology search dialog. The matrix was analyzed until trees of the same minimum length were recovered by 100 replicates of the algorithm, each beginning from a different starting topology. If a particular replicate identified a tree shorter than the existing minimum length trees, the process restarted until 100 replicates had recovered trees of the new minimum length. Novel minimum length trees from each replicate were retained, up to 10,000. Once this process was completed, resulting trees were then submitted for branch swapping in the Traditional Search dialog to ensure that all most parsimonious trees were identified, again with a limit of 10,000 trees in total.

## RESULTS

### Systematic paleontology

MAMMALIA *Linnaeus, 1758*
EUTHERIA *Huxley, 1880*
HYAENODONTA *Van Valen, 1967* (*sensu Solé, 2013*)
HYAENODONTIDAE *Leidy, 1869*
HYAENODONTINAE (*Leidy, 1869*)
*PROPTERODON Martin, 1906*

**Comments**—*Propterodon* was named by *Martin (1906)* without designation of a type species. In 1925, Matthew and Granger named a new species that they referred to *Propterodon*, *P. irdinensis*. In the absence of any prior referral of a species to *Propterodon*,

*P. irdinensis* became, by default, the type species, a situation that spawned considerable taxonomic confusion and was ultimately resolved by *Polly & Lange-Badré (1993)*. *Matthew & Granger (1925)* named *Propterodon irdinensis* based on jaw fragments, not certainly associated, from Inner Mongolian exposures of the middle Eocene Irdin Manha Formation (Irdinmanhan stage) (Fig. 3). The previous year, *Matthew & Granger (1924)* had described *Paracynohyaenodon morrisi* from the same beds, and most recent workers have regarded the two species as conspecific, with *Propterodon morrisi* the appropriate name for this taxon (*Dashzeveg, 1985*; *Polly & Lange-Badré, 1993*; *Morlo & Habersetzer, 1999*). *Dashzeveg (1985)* named an additional hyaenodont taxon, *Pterodon rechetovi*, for two maxillae from the Irdin Manha-equivalent Khaichin Ula 2 fauna from the Khaichin Formation of Mongolia. This species was subsequently made the type species of a new genus, *Neoparapterodon*, by *Lavrov (1996)*, but *Morlo & Habersetzer (1999)*, noting that the upper dentition of *Propterodon morrisi* is essentially identical to that of *N. rechetovi*, placed the latter genus and species in synonymy with the former. In addition to *P. morrisi*, three other species of *Propterodon* have been named. *Propterodon pishigouensis* was named by *Tong & Lei (1986)* for a dentary preserving $P_4$-$M_1$ from the Hetaoyuan Formation (Irdinmanhan), Henan Province, China (Fig. 3). As is discussed below, the affinities of *P. pishigouensis*, do not appear to lie with either *Propterodon* or with Hyaenodonta generally. An additional Chinese species, *P. tongi* was named by *Liu & Huang (2002)* for a dentary with $P_1$-$M_3$ from the Huoshipo locality, Yuli Member of the Hedi Formation (Irdinmanhan), Shanxi Province. This species differs from *P. morrisi* in being slightly smaller and in having a more strongly hypercarnivorous morphology, with metaconids lacking at least on $M_{2-3}$, trigonids more open, and talonids more reduced, especially on $M_3$. Most recently, *Bonis et al. (2018)* named *Propterodon panganensis* for a dentary preserving $P_4$-$M_1$ from the Sharamurunian equivalent Pondaung Formation of Myanmar (Fig. 3). This species has some unusual features (symmetric $P_4$ protoconid, $P_4$ and $M_1$ similar in size, very reduced $M_1$ talonid) that suggest its relationship to other *Propterodon* requires confirmation, but it is clearly a hypercarnivorous hyaenodont.

*PROPTERODON WITTERI*, sp. nov. urn:lsid:zoobank.org:act:4D88F815-E7BE-4997-890F-59BC65A06A28

(Fig. 4, Table 1)

**Holotype**—MCZ VPM 19874, left dentary preserving $M_{2-3}$, the back of the horizontal ramus and almost all of the ascending ramus.

**Etymology**—Named for R. V. Witter, whose party collected the type and only known specimen in 1940.

**Type Locality**—Leota Quarry, Uinta Basin, Uintah County, Utah (Fig. 1B).

**Stratigraphy and Age**—Myton Member of the Uinta Formation (Uinta C, Fig. 1A), late Uintan (Ui$_3$) North American Land Mammal Age (NALMA), late middle Eocene (*Prothero, 1996*) (Fig. 3).

**Diagnosis**—Largest known species of *Propterodon*, with $M_2$ and $M_3$ lengths approximately 11 and 13 mm, respectively, and dentary depth approximately 25 mm beneath $M_3$. Talonid on $M_3$ relatively large, comparable to $M_2$ talonid. Metaconids on $M_{2-3}$ present but extremely reduced.

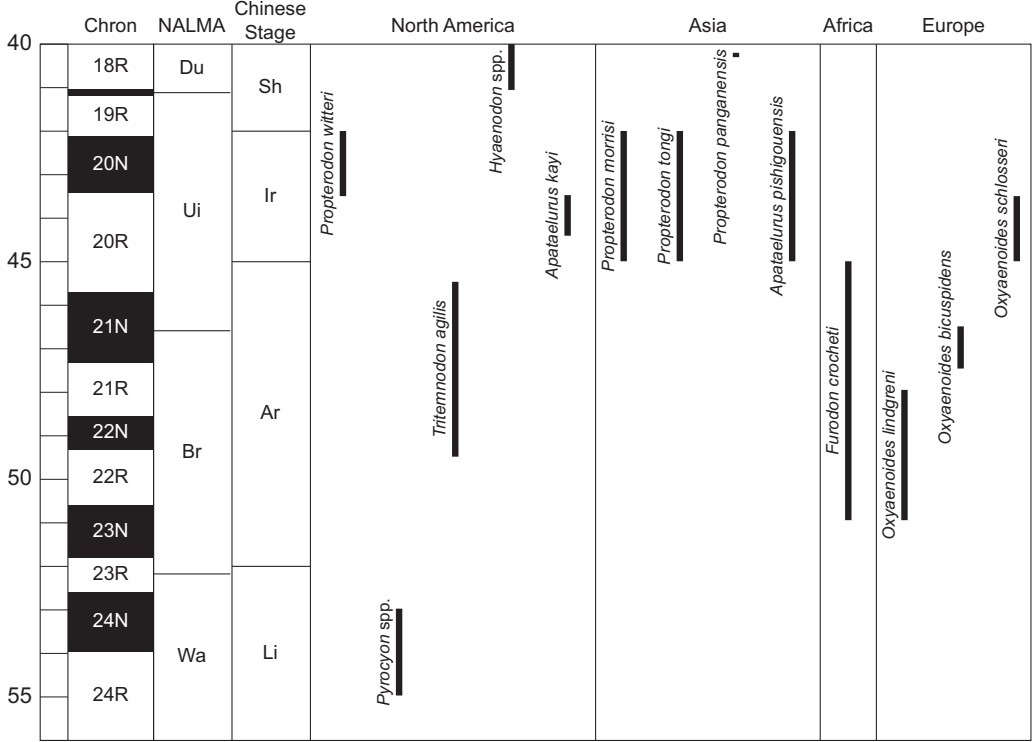

**Figure 3  Temporal distribution of significant taxa discussed in this work.** Geomagnetic polarity chrons follow *Ogg, Ogg & Gradstein (2016)*. North American Land Mammal Age (NALMA) boundaries follow *Tsukui & Clyde (2012)* and *Murphey et al. (2018)*. Chinese stage boundaries follow *Wang et al. (2019)*. Age ranges for hyaenodont and oxyaenodont taxa follow *Prothero (1996)*, *Gunnell et al. (2009)*, *Liu & Huang (2002)*, *Tomiya (2013)*, *Zaw et al. (2014)*, *Solé, Falconnet & Vidalenc (2015)*, *Solé et al. (2016)*, *Wang et al. (2019)*, and personal observation of *Pyrocyon* spp. Abbreviations: Ar, Arshantan; Br, Bridgerian; Du, Duchesnean; Ir, Irdinmanhan; Li, Lingchan; Sh, Sharamurunian; Ui, Uintan; Wa, Wasatchian. Drawings by Shawn P. Zack.

**Differential Diagnosis**—Differs from *P. panganensis* in substantially larger size, with dentary more than 100% deeper. Differs from *P. morrisi* in larger size, approximately 40% longer $M_{2-3}$, more reduced metaconids on $M_{2-3}$, and a relatively larger talonid on $M_3$. Differs from *P. tongi* in larger size, approximately 50% longer $M_{2-3}$, retention of rudimentary metaconids on $M_{2-3}$, larger talonids on $M_{2-3}$, and a less recumbent $M_3$ protoconid.

**Description**—The preserved portion of the horizontal ramus of the dentary is deep and transversely compressed beneath $M_3$ (Figs. 4A–4B). Posterior to the tooth row, the coronoid process forms an approximately 60-degree angle with the alveolar margin. The process is elongate and extends well above the tooth row, although its dorsal extremity is lacking. The posterior margin of the coronoid process is concave, and the process appears to have overhung the mandibular condyle. On the ventral margin of the dentary, there is a slight concavity between the horizontal ramus and the angular process. The angular process itself is directed posteriorly, with no meaningful ventral or medial inflection. The process is relatively thick, with no medial excavation between the angular process and condyle.

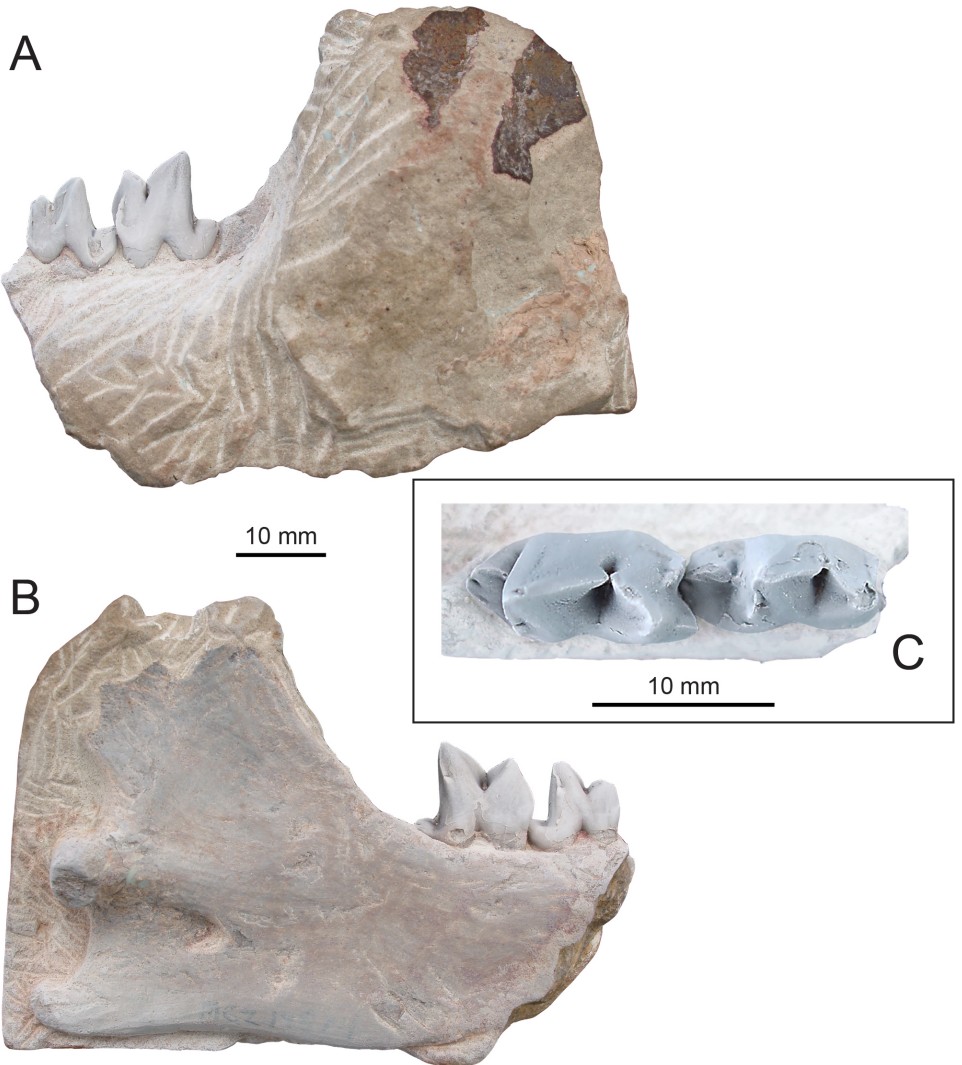

**Figure 4** **Holotype of *Propterodon witteri* sp. nov. (MCZ VPM 19874).** Right dentary with $M_{2-3}$ in (A) buccal, (B) lingual, and (C) occlusal views. Scale bars are 10 mm. Photographs by Shawn P. Zack.

The tip of the process extends posterior to the mandibular condyle and has a slight dorsal curvature. The mandibular condyle is positioned at the level of the alveolar border. The condyle is flush with the ascending ramus, with no development of a neck. The visible portion of the condyle is deepest at its medial margin, tapering dorsolaterally. The bone of the ascending ramus is thickest in a low, broad ridge extending anteriorly and somewhat ventrally from the condyle. Just inferior to this ridge, near mid-length of the ascending ramus is the opening of the mandibular canal.

$M_2$ is complete, aside from slight damage to the apex of the paraconid and the buccal base of the talonid (Figs. 4A–4C). The trigonid is much longer and more than twice the height of the talonid. It would likely have been taller, but a large, vertical wear facet on the buccal surface of the paracristid has removed the apex of the protoconid and likely the

**Table 1** Measurements (mm) of the holotype of *Propterodon witteri.*

| Specimen Number | Locus | L | TrL | TrW | TrH | TaL | TaW | TaH |
|---|---|---|---|---|---|---|---|---|
| MCZ VPM 19874 | $M_2$ | 11.5 | 7.8 | 5.4 | 9.7 | 3.8 | 4.0 | 4.8 |
| | $M_3$ | 13.5 | 10.3 | 6.2 | 12.2 | 3.2 | 3.7 | 4.8 |
| Dentary depth | | 24.7 | | | | | | |

**Notes.**

 Abbreviations as in Fig. 2.

paraconid. The facet extends nearly to the base of the crown and, occlusally, has exposed dentine of both cusps.

The protoconid is the largest and tallest trigonid cusp. The paracristid descends relatively steeply and directly mesially from its apex to meet the paraconid portion of the paracristid in a deep carnassial notch that is continued lingually as a horizontal groove between the paraconid and protoconid. At the distolingual corner of the protoconid, the vertical protocristid is indistinct near the apex of the cusp, becoming better-defined basally and meeting the metaconid in a small carnassial notch.

Mesially, the paraconid is approximately two-thirds the height of the protoconid. The paraconid portion of the paracristid forms an angle of approximately 45 degrees to the long axis of the crown. From its junction with the protoconid portion, it rises slightly towards the paraconid apex. At the mesial margin of the tooth, the paraconid forms a mesial keel that helps define a flattened, diamond-shaped lingual surface. Lingually, the paraconid and protoconid are fused to a level close to three quarters the height of the former cusp. Buccally, the paraconid supports a strong, vertical mesiobuccal cingulid that extends distally, even with the carnassial notch and projects further mesially than the mesial keel. Together, the cingulid and mesial keel form a well-defined embrasure for the back of the talonid of $M_1$.

The metaconid of $M_2$ is a tiny but distinct cusp positioned high on the protoconid, just below the level of the paraconid apex. The metaconid is fused with the protoconid to a level above the level of fusion of the paraconid and protoconid. The apex of the metaconid is directed slightly distally as well as lingually and bears a distinct crest that meets the protoconid portion of the protocristid.

The talonid is dominated by the hypoconid. The apex of the cusp is worn away but was likely flat topped, as in $M_3$. Buccally, the talonid falls away steeply from the apex of the hypoconid and a wear facet occupies most of the buccal surface of the talonid. Lingually, there is a gentler slope, forming a flat, inclined surface. The cristid obliqua is nearly longitudinal in orientation, meeting the base of the trigonid in a small carnassial notch. The contact is buccal to the level of the metaconid, but still well lingual of the buccal margin of the protoconid, resulting in a shallow hypoflexid.

Near the distal margin of the lingual side of the talonid is a shallow groove that appears to separate the hypoconid from a much smaller, lower hypoconulid. There is no entoconid or entocristid. Aside from the mesiobuccal cingulid, there is no development of cingulids. Buccal enamel extends slightly more basally than lingual enamel.

$M_3$ is larger than $M_2$ and almost unworn but is otherwise quite similar in gross morphology (Figs. 4A–4C). The unworn protoconid of $M_3$ is slightly recumbent and the protoconid portion of the paracristid is modestly more elongate than the paraconid portion. The mesial keel of the paraconid is stronger than on $M_2$ and projects further than the mesiobuccal cingulid. The $M_3$ metaconid is even smaller than on $M_2$, reduced to a projection at the end of the almost vertical protocristid. Even in this rudimentary state, a tiny carnassial notch still separates the cusp from the protoconid, but there is no distal projection of the metaconid, unlike $M_2$.

The talonid is shorter than on $M_2$ and, unlike on the latter tooth, is noticeably narrower distally, with its lingual margin running distobuccally from the lingual base of the protoconid. As on $M_2$, the largest cusp on the $M_3$ talonid is the hypoconid. The unworn $M_3$ hypoconid is flat-topped, but the lingual enamel appears to be thickest near its distal margin, indicating a distal position for the hypoconid apex. As on $M_2$, the cristid obliqua meets the trigonid in a small carnassial notch buccal to the level of the metaconid. From that point, the cristid obliqua continues briefly as a vertical crest that ascends the trigonid, reaching approximately one third of the height of the protoconid. The hypoconulid of $M_3$ is small but better defined than on $M_2$, being separated from the hypoconid by a carnassial notch. At the lingual margin of the talonid, opposite the apex of the hypoconid, is a linear thickening of enamel that suggests the presence of a very weak entocristid.

**Comparisons**—The strongly hypercarnivorous morphology of *P. witteri* distinguishes the new species from known Uintan and older North American hyaenodonts. Among named Uintan hyaenodonts (*Matthew, 1899*; *Matthew, 1909*; *Hay, 1902*; *Peterson, 1919*; *Gustafson, 1986*), *Mimocyon longipes* and *Sinopa major* differ dramatically from the new species, with relatively low, closed trigonids, unreduced metaconids, and large, deeply basined talonids. The limnocyonines *Limnocyon potens* and *Oxyaenodon dysodus* show greater carnivorous adaptation than *Mimocyon* or *Sinopa*, but both have more closed trigonids, larger metaconids, and broader, better-developed talonids than *P. witteri*.

Wasatchian *Pyrocyon* and Bridgerian *Tritemnodon* (Fig. 3) more closely approach the morphology of the new species, but with less developed hypercarnivorous adaptation. $M_{2-3}$ in species of *Pyrocyon* (*P. dioctetus*, *P. strenuus*) and in *Tritemnodon agilis* resembles *Propterodon witteri* in having open trigonids (that is, with the paraconid apex well mesial to the apices of the protoconid and, if present, metaconid) with elongate prevallid shearing blades, reduced metaconids, strong mesiobuccal cingulids (particularly in *T. agilis*), small, narrow talonids, and reduced hypoconulids. However, in all of these features, the morphology of *P. witteri* is more extreme, with more open trigonids with more elongate prevallids, much more reduced metaconids, mesiobuccal cingulids that are stronger and more vertical, and more simplified talonids with a very weak to absent entoconid/entocristid complex, which is retained in both *Pyrocyon* and *Tritemnodon*. In addition, in both *Pyrocyon* and *Tritemnodon*, $M_3$ is subequal to $M_2$, while in *P. witteri*, it is substantially larger. *Tritemnodon agilis* further differs from *P. witteri* in having a shallower, more gracile dentary and a more inclined (less vertical) coronoid process.

The temporal gap between *Propterodon witteri* and species of *Pyrocyon* and *Tritemnodon* is also problematic (Fig. 3). *Pyrocyon* is well-known known from mid-Wasatchian faunas (*Gingerich & Deutsch, 1989*) but does not appear to persist until the end of the interval. In the Willwood Formation of the Bighorn Basin, *Pyrocyon* disappears from the record during $Wa_6$, well before the end of the densely sampled portion of the Willwood record (*Chew, 2009*), and the genus is unknown from $Wa_7$ through Uintan faunas. *Tritemnodon* is well-documented from the earlier portion of the Bridgerian, particularly $Br_2$, but has a limited record from $Br_3$ and no record from the earlier portions of the Uintan ($Ui_{1-2}$) (*Eaton, 1982*; *Gunnell et al., 2009*). A close relationship of *P. witteri* to either genus would imply substantial gaps in the hyaenodont record.

Hypercarnivorous hyaenodonts are also present in mid-Eocene faunas from Africa (*Furodon*), Asia (*Propterodon*), and Europe (*Oxyaenoides*) (*Matthew & Granger, 1924*; *Matthew & Granger, 1925*; *Lange-Badré & Haubold, 1990*; *Lavrov, 1996*; *Liu & Huang, 2002*; *Solé et al., 2014*; *Solé, Falconnet & Vidalenc, 2015*; *Solé et al., 2016*; *Godinot et al., 2018*) (Fig. 3). Unlike *Pyrocyon* or *Tritemnodon*, $M_3$ is distinctly larger than $M_2$ in these taxa, a similarity shared with *P. witteri*. A link to one or more of these taxa would have implications for the origins of the Uinta form and for intercontinental dispersals of hyaenodonts more generally.

Compared to *Propterodon witteri* the $M_{2-3}$ trigonids of species of European *Oxyaenoides* (*O. bicuspidens*, *O. lindgreni*, *O. schlosseri*) are more closed, with a shorter paraconid portion of the paracristid (*Lange-Badré & Haubold, 1990*; *Solé, Falconnet & Yves, 2014*; *Solé, Falconnet & Vidalenc, 2015*; *Godinot et al., 2018*) (Figs. 5C–5D). *Oxyaenoides* has completely lost metaconids on all molars, while *P. witteri* retains small metaconids on $M_{2-3}$. In *Oxyaenoides*, the protoconid and paraconid are separated to a level close to the base of the crown, contrasting with *P. witteri*, where these cusps are fused to approximately mid-height. Both taxa have a distinct mesiobuccal cingulid, but it is much lower in *Oxyaenoides*. While both have reduced talonids, the hypoconulid is relatively larger in *Oxyaenoides* and a more distinct entoconid/entocristid complex is retained, even in the derived *O. schlosseri*. *Oxyaenoides* talonids are also much shorter relative to their width than in *P. witteri*. Overall, *Propterodon witteri* displays a mixture of more derived morphologies (open trigonids, trenchant talonids) and less derived morphologies (retained metaconids, elongate talonids) in comparison to *Oxyaenoides*. This pattern is suggestive of parallel developments in lineages assembling a hypercarnivorous morphology independently.

African *Furodon crocheti* has more closed trigonids than *Propterodon witteri* (*Solé et al., 2014*) (Figs. 5E–5F). However, the length of the paraconid portion of the prevallid blade is similar, resulting in the paraconid overhanging the lingual margin of the crown in *F. crocheti*. The metaconid is larger in *F. crocheti* than in *P. witteri*. However, whereas in *P. witteri*, the metaconid is positioned high on the protoconid, almost at the same height as the paraconid apex, it is positioned much lower in *F. crocheti*. As a result, despite its size, the metaconid apex is substantially lower than the paraconid apex. The talonids of *F. crocheti* are relatively larger than in *P. witteri*, particularly on $M_2$, and the $M_2$ talonid is much wider as well. The $M_2$ hypoconid has a mesial apex in *F. crocheti*, with a subequal cristid obliqua and hypocristid. In *P. witteri*, the apex of the hypoconid is distal and there

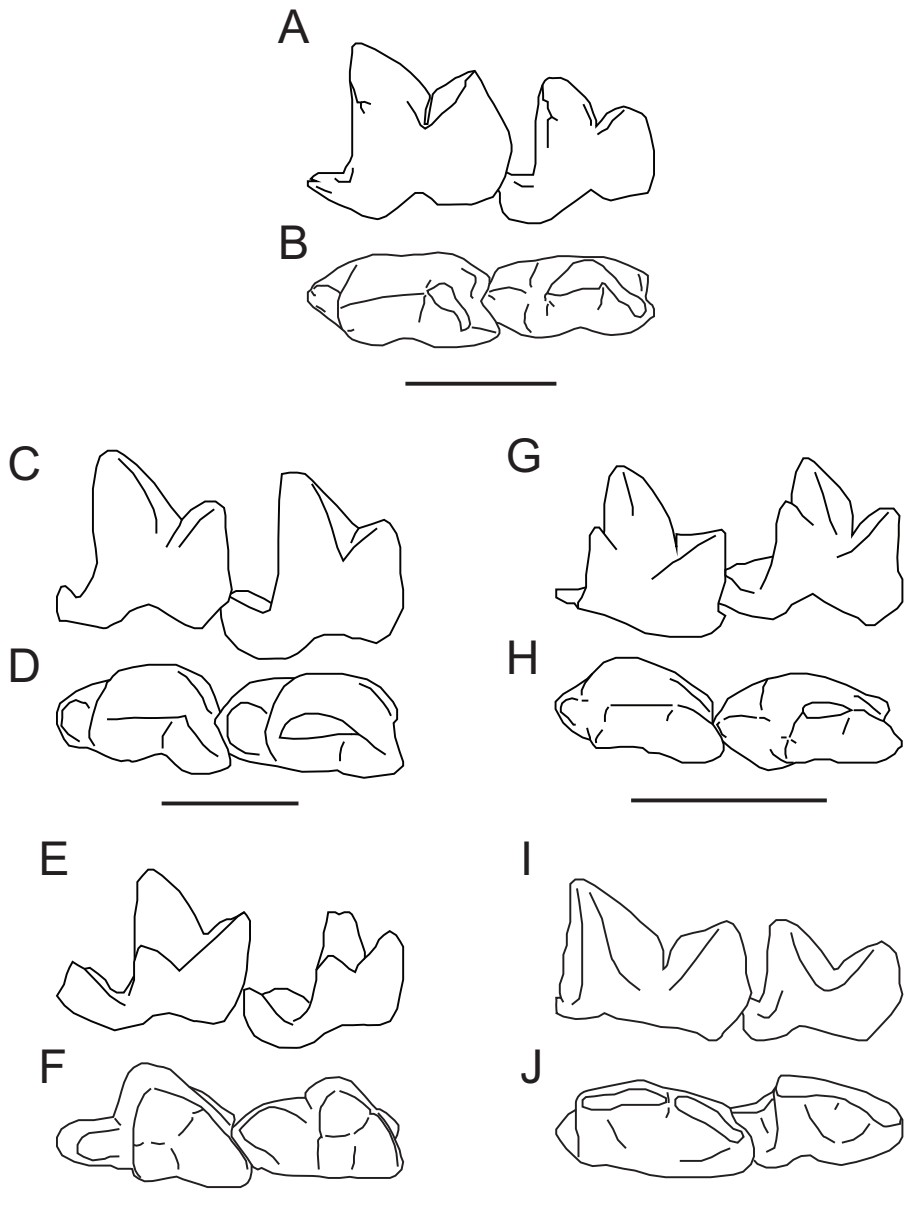

**Figure 5** **Comparison of M$_{2-3}$ of *Propterodon witteri* sp. nov. with other middle Eocene hypercarniv-orous hyaenodonts.** Left M$_{2-3}$ of *Propterodon witteri*, MCZ VPM 19874, in (A) lingual and (B) occlusal views. Right M$_{2-3}$ (reversed) of *Oxyaenoides schlosseri*, MNHN.F.ERH 429, in (C) lingual and (D) occlusal views. Left M$_{2-3}$ of *Furodon crocheti*, HGL 50bis-56, in (E) lingual and (F) occlusal views. Right M$_{2-3}$ (reversed) of *Propterodon morrisi*, AMNH FM 21553, in (G) lingual and (H) occlusal views. Left M$_{2-3}$ of *Propterodon tongi*, IVPP V12612, in (I) lingual and (J) occlusal views. All scale bars are 10 mm. Drawings by Shawn P. Zack. (A–B) and (G–H) drawn from photographs by Shawn P. Zack. (C-D) drawn from *Solé, Falconnet & Vidalenc* (*2015*, fig. 4). (E–F) drawn from *Solé et al.* (*2014*, fig. 2). (I–J) drawn from photographs provided by M. Borths.

is no hypocristid to speak of. While the hypoconulid appears to be small in *F. crocheti*, the entoconid/entocristid complex remains prominent, contrasting with the trenchant morphology present in *P. witteri*. Finally, on the dentary of *F. crocheti*, the ventral margin of the angular process grades smoothly into the horizontal ramus, lacking the distinct inflection that occurs in *P. witteri*.

Some of the features that distinguish *F. crocheti* from *P. witteri* are shared with other, less hypercarnivorous taxa from Africa and South Asia. The paraconid overhang is present in African *Brychotherium* and South Asian Indohyaenodontinae (*Kumar, 1992*; *Egi et al., 2005*; *Rana et al., 2015*; *Borths, Holroyd & Seiffert, 2016*), while the low placement of the metaconid is shared with these taxa as well as African *Glibzegdouia* and *Masrasector Solé et al., 2014*; *Borths & Seiffert, 2017*). A mesially positioned hypoconid apex occurs in *Glibzegdouia*, *Masrasector*, and the indohyaenodontines *Kyawdawia* and *Yarshea* (*Egi et al., 2004*; *Egi et al., 2005*; *Solé et al., 2014*; *Borths & Seiffert, 2017*). These similarities are consistent with phylogenetic analyses that link *Furodon* to African and South Asian hyaenodonts (*Rana et al., 2015*; *Borths, Holroyd & Seiffert, 2016*; *Borths & Seiffert, 2017*; *Borths & Stevens, 2019a*; *Borths & Stevens, 2019b*). Their absence in *Propterodon witteri* indicate that its affinities lie elsewhere.

The morphology of the two best known species of Asian *Propterodon*, *P. morrisi* (senior synonym of the type species, *P. irdinensis*) (Figs. 5G–5H) and *P. tongi* (Figs. 5I–5J), is quite similar to that of *P. witteri* (*Matthew & Granger, 1924*; *Matthew & Granger, 1925*; *Liu & Huang, 2002*). Trigonid proportions of $M_{2-3}$ in *P. morrisi* (e.g., AMNH FM 21553) are nearly identical to *P. witteri*, while *P. tongi* has slightly more open trigonids than either species. In *P. morrisi*, the metaconids of $M_{2-3}$ are reduced but remain slightly larger than in *P. witteri*. The opposite is true of *P. tongi*, with both $M_2$ and $M_3$ lacking defined metaconids. In *P. morrisi*, the metaconids are positioned high on the protoconid, comparable to *P. witteri*. Both Asian species have well-developed, vertical mesiobuccal cingulids that extend high up on the paraconid. Talonid structure is also closely comparable, at least on $M_2$. The Asian species have small talonids (smaller in *P. tongi*) with distal hypoconid apices, rudimentary hypoconulids positioned directly distal to the hypoconid, and no entoconid/entocristid complex, all identical to the morphology on $M_2$ of *P. witteri*. The $M_3$ talonid is more reduced in the Asian forms than in the North American taxon. In the case of *P. tongi*, it is reduced to a cuspule on the distal end of the trigonid. The talonid is larger in *P. morrisi*, but still smaller than in *P. witteri*. As in the North American form, there does appear to be a trace of an entocristid on the $M_3$'s of AMNH FM 20128 and 21553. Taken together, the morphology of *Propterodon witteri* is closely comparable to *P. morrisi* and *P. tongi*, particularly the former. The most significant morphological distinction is the relative size of the $M_3$ talonid, which is relatively larger in *P. witteri* than in either Asian species. Despite this contrast, Asian *Propterodon* species are clearly the closest matches to *P. witteri* among relevant taxa, and referral of the new species to *Propterodon* can be made with confidence.

**Phylogenetic Results**—Analysis of the matrix described in Materials & Methods produced 145 most parsimonious trees ($L = 510$, CI $= 0.294$, RI $= 0.615$), the majority rules consensus of which is shown in Fig. 6. Resolution is poor, even using the

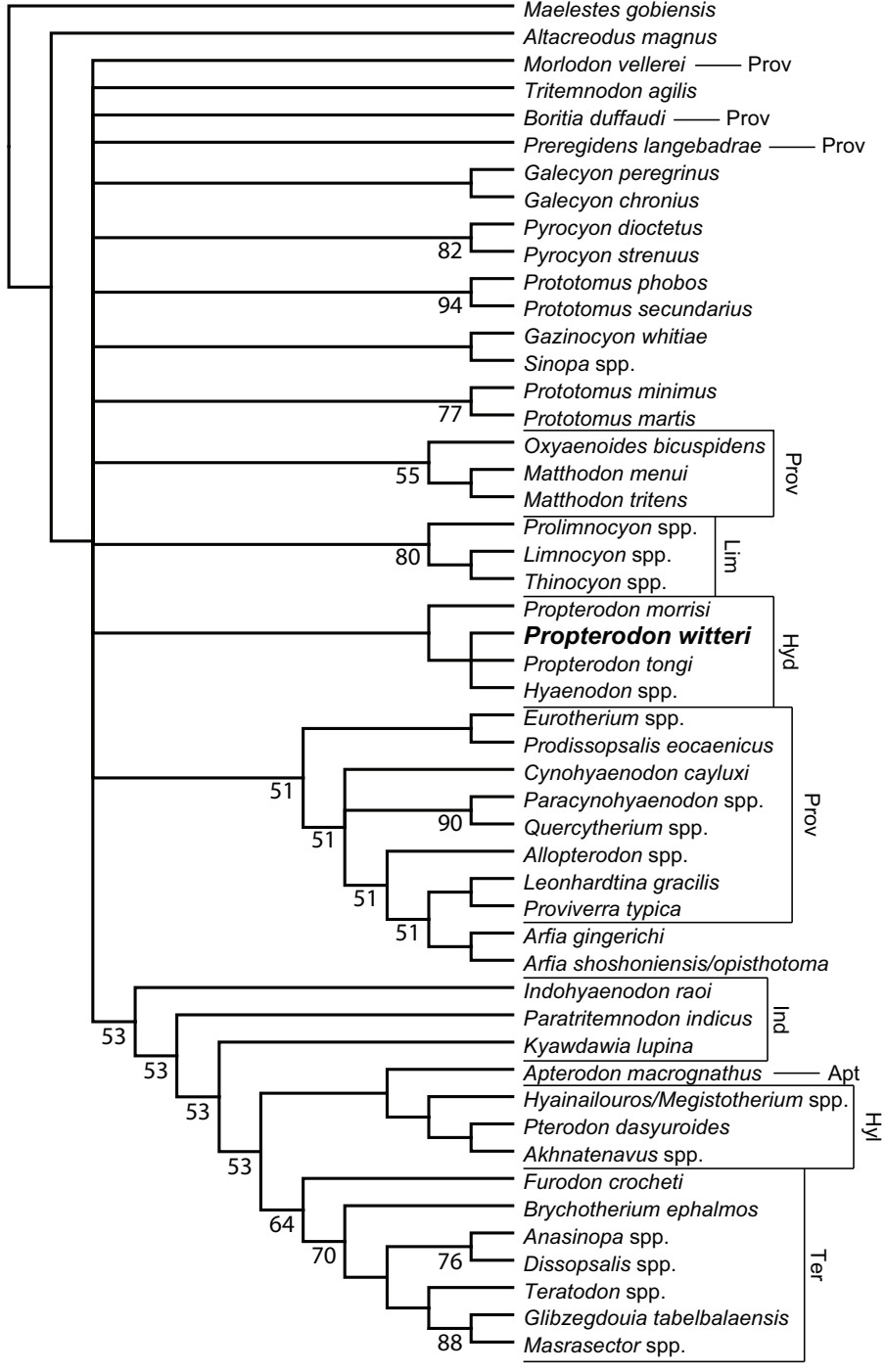

**Figure 6** **Phylogenetic position of *Propterodon witteri* sp. nov.** Majority rule consensus of 145 most parsimonious trees (L = 510, CI = 0.294, RI = 0.615) showing the inferred phylogenetic position of *Propterodon witteri* sp. nov. Numbers below branches indicate percent support, where less than 100 percent. Subfamilies mentioned in the text are labelled. Taxa included in Proviverrinae follows *Solé et al. (2015)*. Abbreviations: Apt, Apterodontinae; Hyd, Hyaenodontinae; Hyl, Hyainailourinae; Ind, Indohyaenodontinae; Lim, Limnocyoninae; Prov, Proviverrinae; Ter, Teratodontinae. Drawings by Shawn P. Zack.

majority rules rather than a strict consensus. The largest clade unites a paraphyletic Indohyaenodontinae with the three primary African subfamilies (Hyainailourinae, Apterodontinae, Teratodontinae). A second major clade comprises most members of Proviverrinae along with *Arfia*, which is unexpectedly deeply nested within Proviverrinae as the sister taxon of *Proviverra* and *Leonhardtina*. Smaller groupings include Limnocyoninae, Hyaenodontinae, and groupings of the North American *Sinopa* and *Gazinocyon* and the European hypercarnivorous genera *Oxyaenoides* and *Matthodon*. All of these clades form a massive polytomy at the base of the ingroup, along with numerous genera and species of early and middle Eocene hyaenodont.

While disappointing, the poor resolution of the consensus tree is consistent with a lack of clarity in other recent analyses of hyaenodont phylogeny. While the consensus topology is better resolved, most clades recovered by *Rana et al. (2015)* have poor bootstrap support. This is also true in other recent analyses using parsimony (*Borths, Holroyd & Seiffert, 2016*; *Borths & Seiffert, 2017*). Most nodes in Bayesian trees recovered by Borths and colleagues (*Borths, Holroyd & Seiffert, 2016*; *Borths & Seiffert, 2017*; *Borths & Stevens, 2017*; *Borths & Stevens, 2019a*; *Borths & Stevens, 2019b*) have similarly low posterior probabilities, and there are substantial topological differences between analyses with different assumptions concerning character evolution (e.g., Prionogalidae in *Borths & Stevens, 2019a*, supplementary fig. 1 versus 2). Simply put, many relationships within Hyaenodonta are neither stable nor well-resolved.

With regard to *Propterodon witteri*, two conclusions can be made. First, all trees recover a clade linking the new species to *Propterodon morrisi*, *P. tongi*, and *Hyaenodon*. Monophyly of *Propterodon* is not recovered, with a majority of trees linking *P. tongi* and *P. witteri* more closely to *Hyaenodon* than to *P. morrisi* on the basis of greater metaconid and entoconid reduction in the former species. These results indicate that *Propterodon* is paraphyletic and is likely to be directly ancestral to *Hyaenodon*, although further support would be desirable, particularly as metaconid and entoconid reduction have occurred convergently in many different lineages of carnivorous mammal (e.g., *Muizon & Lange-Badré, 1997*).

In addition, the position of Hyaenodontinae within Hyaenodonta is not well-resolved. While hyaenodontine monophyly is supported in all shortest trees, the subfamily is recovered in the large polytomy at the base of the ingroup. This contrasts with recent analyses that have consistently supported some form of a link to European hyaenodonts (*Rana et al., 2015*; *Borths, Holroyd & Seiffert, 2016*; *Borths & Seiffert, 2017*; *Solé & Mennecart, 2019*; *Borths & Stevens, 2019a*; *Borths & Stevens, 2019b*), particularly the hypercarnivorous *Oxyaenoides*. The implications of this aspect of the topology are discussed below

One other result that warrants brief comment is that the two recently described European hyaenodont genera, both described as potential proviverrines (*Solé, Falconnet & Yves, 2014*; *Solé, Falconnet & Vidalenc, 2015*), *Boritia* and *Preregidens*, are not recovered in proximity to Proviverrinae. Instead, many individual trees recover these genera in positions proximate to species of *Prototomus* (specifically *P. martis* and *P. minimus*) and *Pyrocyon*. This includes trees in which the European genera are successive sister taxa to *Pyrocyon* and trees in which *Preregidens* is the sister taxon of *Prototomus minimus* (with *P. martis* as sister taxon to

this clade). Consistent with this result, both genera lack the distinctive enlarged, bulbous entoconid typical of proviverrine molar talonids (e.g., *Solé, 2013*). Of the two, *Boritia* is very similar to several early Eocene North American hyaenodonts (*Prototomus martis*, *Pyrocyon* spp.), and it may represent a parallel development from an early European species of *Prototomus* (e.g., *P. girardoti*). Alternatively, it may document evidence of faunal exchange between North America and Europe after the Paleocene-Eocene Thermal Maximum, consistent with evidence from the Abbey Wood fauna (*Hooker, 2010*).

OXYAENODONTA *Van Valen, 1971*
OXYAENIDAE *Cope, 1877*
MACHAEROIDINAE *Matthew, 1909*
*APATAELURUS Scott, 1937*
*APATAELURUS PISHIGOUENSIS Tong & Lei, 1986*, comb. nov.
(Fig. 7)
*?Propterodon pishigouensis Tong & Lei, 1986*:212, Fig. 2, pl. 1.3
*?Propterodon shipigouensis Tong, 1997*:6 (lapsus calami)

**Holotype**—IVPP V7997, left dentary preserving $P_4$-$M_1$.
**Type Locality**—Shipigou, Liguanqiao Basin, Xichuan County, Henan Province, China.
**Stratigraphy and Age**—Hetaoyuan Formation, Irdinmanhan stage (*Wang et al., 2019*).
**Revised Diagnosis**—Smallest known species of *Apataelurus*, with $P_4$ and $M_1$ lengths approximately 10 and 9 mm, respectively.
**Comparisons and Discussion**—*Tong & Lei (1986)* described IVPP V7997 as a new species of *Propterodon*, *P. pishigouensis*. Compared to other species referred to *Propterodon*, the most distinctive feature of "*P*". *pishigouensis* is the shape of the dentary, which is ventrally deflected anteriorly, beginning below the anterior root of $P_4$ (*Tong & Lei, 1986*), indicating the presence of an anterior flange (Fig. 7A). In contrast, the symphysial region is shallow in *P. morrisi* and *P. tongi* and tapers anteriorly. In fact, an anterior dentary flange has not been documented in any hyaenodont. The only middle Eocene carnivorous mammals known to possess such a flange are machaeroidines (*Scott, 1938*; *Matthew, 1909*; *Gazin, 1946*; *Dawson et al., 1986*), a small clade of North American Wasatchian through Uintan carnivores recently supported as oxyaenids (*Zack, 2019*).

Machaeroidines, particularly the Uintan *Apataelurus kayi*, share substantial similarities with the type specimen of "*Propterodon*" *pishigouensis*, including features that distinguish the latter species from other *Propterodon* (Fig. 7). On $P_4$, both *A. kayi* and *pishigouensis* have a well-developed paraconid that is nearly as tall as the talonid (*Scott, 1938*; *Tong & Lei, 1986*). The paraconid is absent on $P_4$ in *P. tongi* (*Liu & Huang, 2002*). In *P. panganensis* it is low and weakly developed (*Bonis et al., 2018*). While all relevant species have simple $P_4$ talonids dominated by a tall hypoconid, in *pishigouensis* and *A. kayi*, the talonid is distinctly broader than the remainder of the crown (*Scott, 1938*; *Tong & Lei, 1986*). In contrast, $P_4$ width is uniformly narrow in *P. panganensis* and *P. tongi* (*Liu & Huang, 2002*; *Bonis et al., 2018*). In *Propterodon tongi* and, to judge the roots of $P_4$, *P. morrisi*, $P_4$ is enlarged relative

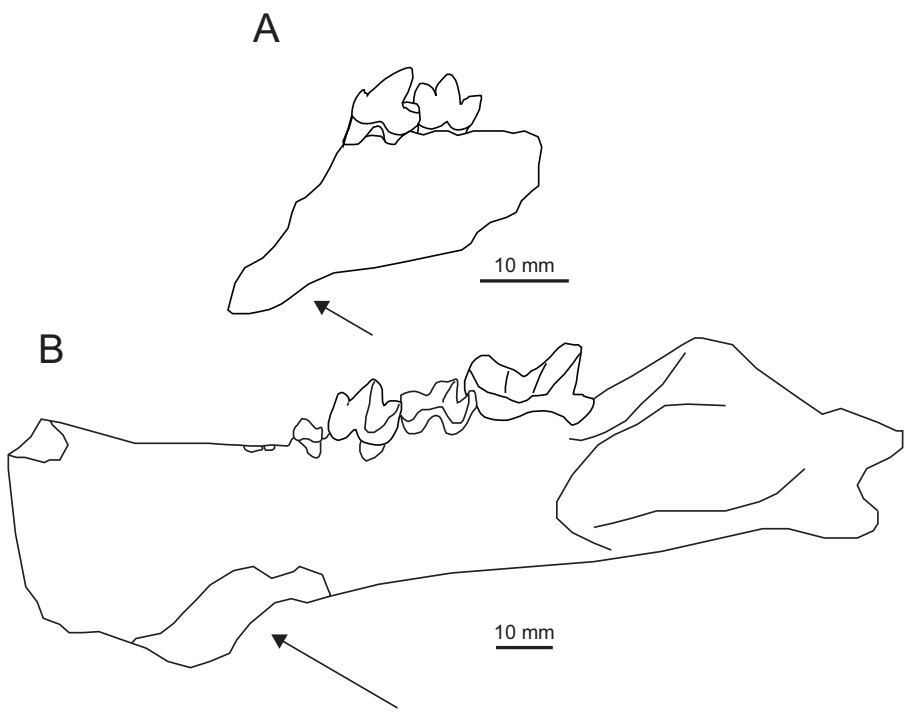

**Figure 7 Comparison of *Apataelurus pishigouensis* comb. nov. with *A. kayi*.** (A) *Apataelurus pishigouensis*, IVPP V7997, left dentary with P$_4$-M$_1$; (B) *Apataelurus kayi*, CM 11920, right dentary with P$_3$-M$_2$ (reversed). Both images show the dentary in buccal view. Arrows indicate the ventral deflection of the dentaries of both specimens. Note that the apparently greater height of the protoconids on P$_4$ and M$_1$ and paraconid on M$_1$ in *A. pishigouensis* reflects much heavier wear in *A. kayi*. All scale bars are 10 mm. Drawings by Shawn P. Zack. (A) drawn from *Tong & Lei* (*1986*, pl. 1). (B) drawn from a photograph by Shawn P. Zack.

to M$_1$ (*Matthew & Granger, 1925*; *Liu & Huang, 2002*). In *pishigouensis* and *A. kayi*, along with *P. panganensis*, the two teeth are subequal in size (*Scott, 1938*; *Tong & Lei, 1986*; *Bonis et al., 2018*).

On M$_1$, a defined metaconid is lacking in *pishigouensis* and *A. kayi* (*Scott, 1938*; *Tong & Lei, 1986*), again along with *P. panganensis* (*Bonis et al., 2018*), but retained in *P. morrisi* (e.g., AMNH FM 21553), with M$_1$ of *P. tongi* too worn to assess. The primary difference in M$_1$ morphology is in the talonid. The talonids of *P. morrisi*, *P. tongi*, and *P. panganensis* are short and much lower than the paraconid (*Matthew & Granger, 1925*; *Liu & Huang, 2002*; *Bonis et al., 2018*; pers. obs. of AMNH FM 21553). In *pishigouensis* and *A. kayi*, the talonid is relatively elongate and nearly as tall as the paraconid (*Scott, 1938*; *Tong & Lei, 1986*). Talonid morphology is simplified in both *pishigouensis* and *A. kayi*, with both taxa only retaining a hypoconid. In *P. morrisi* and *P. tongi*, some lingual structure is retained, although the extremely reduced talonid of *P. panganensis* is also simplified.

Taken together, the mandibular and dental morphology of "*Propterodon*" *pishigouensis* differs substantially from other species of *Propterodon*, particularly *P. morrisi* and *P. tongi*, but closely matches the morphology of the North American machaeroidine *Apataelurus kayi*. Accordingly, *Propterodon pishigouensis* is recombined as *Apataelurus pishigouensis*.

As a species of *Apataelurus*, *A. pishigouensis* differs from *A. kayi* primarily in its somewhat smaller size. The talonid of *A. pishigouensis* may be smaller than that of *A. kayi*, but this is complicated by heavier wear in the type and only described specimen of the North American form. Referral of *pishigouensis* to Machaeroidinae represents the first clear record of a machaeroidine in Asia.

There may be an additional, older Asian machaeroidine, also initially described as a hyaenodont. *Isphanatherium ferganensis* was named for an isolated upper molar from the Andarak-2 fauna (*Lavrov & Averianov, 1998*). The morphology of *I. ferganensis* is strikingly derived for an early hyaenodont, with an extremely elongate, longitudinally oriented postvallum blade and a strongly reduced protocone. Both of these features would be consistent with a machaeroidine identity. The overall morphology of the type of *I. ferganensis* is closely comparable to $M^1$ of *Machaeroides* spp. from the early and middle Eocene of North America (*Gazin, 1946*; *Dawson et al., 1986*). They share development and orientation of the metastylar blade, protocone reduction without mesiodistal compression, fusion of the paracone and metacone to a point close to their apices, with the metacone taller than the paracone, and the presence of a low but distinct parastyle that is continuous with a buccal cingulum that is restricted to the mesial portion of the crown. A specific similarity shared by *I. ferganensis* and *M. simpsoni* (S Zack, pers. obs., 2019 of CM 45115) is the presence of contrasting compression of the paracone and metacone, with the former compressed mesiodistally while the latter is compressed transversely. More material is needed to be certain, but the age and morphology of *Isphanatherium ferganensis* supports the tentative reidentification of the species as a machaeroidine and of the holotype as an $M^1$ rather than an $M^2$.

## DISCUSSION

**Hyaenodontine Origins**—Recent assessments of hyaenodont biogeography (*Borths, Holroyd & Seiffert, 2016*; *Borths & Stevens, 2017*) have supported a European divergence of Hyaenodontinae from *Oxyaenoides*, which was recovered as the sister taxon of Hyaenodontinae in both analyses. This grouping is nested within a broader assemblage of European hyaenodonts comprising taxa referred to Proviverrinae by *Solé (2013)* and *Solé et al. (2015)*. More recent studies (*Borths & Stevens, 2019a*; *Borths & Stevens, 2019b*; *Solé & Mennecart, 2019*) complicate this scenario slightly by recovering Prionogalidae and *Thereutherium* within the clade defined by *Oxyaenoides* and Hyaenodontinae, but the basic biogeographic scenario is unchanged, with Hyaenodontinae deeply nested within a clade of European hyaenodonts. As was noted by *Borths & Stevens (2019a)* with regard to the position of Prionogalidae, the character support uniting *Oxyaenoides*, *Thereutherium*, Prionogalidae, and Hyaenodontinae consists primarily of features associated with hypercarnivory, specifically reduction of the metaconids and talonids on lower molariform teeth. Hypercarnivory has evolved iteratively in diverse carnivorous mammalian clades and homoplasy in features associated with hypercarnivory is well-documented (*Muizon & Lange-Badré, 1997*; *Holliday & Steppan, 2004*; *Solé & Ladevèze, 2017*). Accordingly, support for a close relationship between *Oxyaenoides* and Hyaenodontinae should be regarded cautiously, despite its recovery in several analyses.

In contrast to the analyses just discussed, results of the current phylogenetic analysis do not place Hyaenodontinae phylogenetically proximate to *Oxyaenoides*, nor do the results of the *Rana et al. (2015)* analysis. While the position of Hyaenodontinae is not consistently resolved in the present study, a sister taxon relationship to *Oxyaenoides* is not present in any most parsimonious tree. Some most parsimonious trees (MPTs) do recover Hyaenodontinae as the sister taxon of Proviverrinae, as used by *Solé (2013)* and *Solé et al. (2015b)*. However, other MPTs recover Hyaenodontinae as the sister taxon of North American and European *Galecyon* or to a clade comprising *Galecyon* plus Holarctic *Arfia*. Still other MPTs place Hyaenodontinae at the base of a diverse grouping that includes all sampled taxa excepting *Arfia* and Proviverrinae, with Asian and North American Limnocyoninae the next diverging clade. There is no particular support in this analysis for a European origin for Hyaenodontinae.

In fact, a European origin appears unlikely. Unlike *Oxyaenoides*, which shares some distinctive dental features with other proviverrines, including a double-rooted $P_1$ and molar talonids with three, more or less equally developed and equidistantly spaced cusps, hyaenodontine dental morphology has little in common with proviverrines. The relatively large $P_1$ remains single-rooted in *P. morrisi* and *P. tongi* (*Matthew & Granger, 1925*; *Liu & Huang, 2002*), while the entoconid and hypoconulid are weakly developed in all species of *Propterodon*. With the exception of a reduced metacingulum on $M^{1-2}$, other distinctive proviverrine dental features enumerated by *Solé (2013)* (entoconids on $P_{3-4}$, prominent paraconids on $P_{2-3}$ and parastyle on $P^4$, $M^{1-2}$ with metacones taller than paracones) are absent in *Propterodon* (*Matthew & Granger, 1925*; *Lavrov, 1996*; *Liu & Huang, 2002*).

Biogeographic evidence also suggests that derivation of hyaenodontines from within the European Eocene hyaenodont radiation is unlikely. From the late early Eocene through the Eocene/Oligocene transition, Europe was an island isolated from the rest of Holarctica (e.g., *Meulenkamp & Sissingh, 2003*), resulting in the evolution of a diverse endemic mammalian fauna (*Hooker, 1989*; *Badiola et al., 2009*; *Danilo et al., 2013*). This period encompasses the radiation of proviverrine hyaenodonts (sensu *Solé, 2013*), which formed the dominant carnivorous element of this endemic European fauna. There is little evidence of mammalian dispersal from Europe to Asia during this interval.

In fact, there is some evidence from the fossil record consistent with an earlier Asian record of Hyaenodontinae. The ?Arshantan fauna from Andarak-2, Khaichin Formation, Kyrgyzstan, includes a fragmentary hyaenodont dentition (ZIN 34494) described by *Lavrov & Averianov (1998)* as similar to *Neoparapterodon rechetovi*, the latter a likely synonym of *Propterodon morrisi* according to *Morlo & Habersetzer (1999)*. If correctly identified, this would extend the Asian record of Hyaenodontinae back to the early part of the middle Eocene and would support an Asian origin for the subfamily. Unfortunately, the hyaenodont record from both the Arshantan and the preceding Lingchan (equivalent to the Bumbanian) is very poor. Aside from ZIN 34494, the published hyaenodont record from the Arshantan is limited to the type specimen of *Isphanatherium ferganensis* (*Lavrov & Averianov, 1998*), which may not be a hyaenodont (see above). Lingchan hyaenodont records comprise two specimens referred to distinct species of *Arfia* and two specimens referred to *?Prototomus* sp. (*Lavrov & Lopatin, 2004*; *Tong & Wang, 2006*; *Morlo et al., 2014*;

*Solé, Gheerbrant & Godinot, 2013*). Until early and early middle Eocene hyaenodonts from Asia are better documented, it is difficult to determine what role, if any, Asia played in the origin of Hyaenodontinae.

**Late Uintan Carnivore Dispersals**—In addition to *Propterodon*, several other carnivorous taxa that first appear in the late Uintan ($Ui_{2-3}$) have a potential origin outside western North America. Among hyaenodonts, the limnocyonine *Oxyaenodon dysodus* is quite distinct from *Limnocyon potens*, the only limnocyonine known from the early Uintan. Compared to *L. potens*, *O. dysodus* is smaller and more hypercarnivorously adapted, with smaller, less basined talonids and a longer $M_2$ prevallid blade. *Oxyaenodon dysodus* also retains a full complement of relatively uniform incisors, while *L. potens* has enlarged $I^2$ and lost $I^3$ (*Denison, 1938*). While *Morlo & Gunnell (2005)* recovered *O. dysodus* and *L. potens* as sister taxa in a phylogenetic analysis of limnocyonines, an earlier analysis of a nearly identical matrix (*Morlo & Gunnell, 2003*) recovered *O. dysodus* as the sister taxon of Bridgerian *Thinocyon medius*, outside of a monophyletic *Limnocyon* (note that the consensus tree shown in *Morlo & Gunnell* (*2005*, fig. 1) is in error; all four shortest trees found by analyzing the published matrix without modification recover *Thinocyon medius* rather than Bridgerian *Limnocyon* as the sister taxon of *L. potens* plus *O. dysodus*). Both *Morlo & Gunnell (2003)* and *Tong & Lei (1986)* have noted similarities to the Irdinmanhan Chinese taxon *Prolaena parva*. Taken together, it is plausible that the appearance of *Oxyaenodon* in the late Uintan reflects immigration from Asia, similar to the pattern hypothesized for *P. witteri*. A full assessment of the affinities of *Oxyaenodon* is beyond the scope of this study. Published descriptions and illustrations of material of *O. dysodus* are inadequate to confidently score the species, and substantial additional material remains unpublished (*Friscia & Dunn, 2016*).

The affinities of another late Uintan hyaenodont, the small undescribed taxon or taxa referenced above are unclear at present, but small hyaenodontid material from the Mission Valley Formation appear to document a non-limnocyonine with a narrow $M_1$ talonid (S Zack, pers. obs., 2019), very divergent from both *Limnocyon* or *Sinopa*, the only hyaenodont genera known from the early Uintan.

Other carnivorous groups show a similar pattern. At least two machaeroidine taxa are present in late Uintan faunas (*Scott, 1937*; *Scott, 1938*; *Rasmussen et al., 1999*; *Wagner, 1999*; *Zack, 2019*), but none is known from $Ui_1$. Among miacids, several taxa appear in the late Uintan without obvious $Ui_1$ antecedents, including *Tapocyon* spp., *"Miacis" uintensis*, and *"M." hookwayi* (*Wesley & Flynn, 2003*; *Spaulding & Flynn, 2009*; *Tomiya, 2013*). Finally, the enigmatic carnivorous mammal *Simidectes* first appears in the late Uintan, again without obvious early Uintan relatives (*Coombs, 1971*).

The lack of an early Uintan ancestry for some taxa may reflect limited data from the $Ui_1$ interval, which remains relatively poorly sampled. With this caveat, the discovery of *Propterodon witteri* is evidence of a potential Asian origin for many of the carnivorous taxa that first appear in the late Uintan. Referral of *Propterodon pishigouensis* to *Apataelurus* documents an additional tie between the carnivorous faunas of the Irdinmanhan and Uintan. In addition, both the hyaenodont *Sinopa* and the mesonychid *Harpagolestes* are shared by Irdinmanhan and Uintan faunas (*Jin, 2005*; *Jin, 2012*; *Morlo et al., 2014*;

*Robson et al., 2019*). The Huadian Formation fauna containing *S. jilinia* was considered post-Irdinmanhan in age by *Morlo et al. (2014)* based on the stage of evolution of the omomyid *Asiomomys*, but the presence of *Zelomys*, a genus otherwise known from the Irdinmanhan Yuli Member of the Hedi Formation (*Dawson et al., 2003*) suggests an older age. Carnivore dispersals from Asia to North America during the later Uintan would be concordant with evidence for dispersal of other mammals from Asia to North America during this interval, including the chalicotheroid perissodactyl *Grangeria* and the omomyid primate *Macrotarsius* in $Ui_2$ (*Woodburne, 2004*). $Ui_3$ sees additional dispersals including several brontotheriid perissodactyls, and *Mytonolagus*, the oldest known North American lagomorph (*Woodburne, 2004*; *Mihlbachler, 2008*).

A complicating factor is the poor quality of the Asian middle Eocene carnivore record. As discussed above, the Lingchan and Arshantan record of hyaenodonts is extremely poor, and other carnivorous clades are also poorly sampled in both intervals. The Irdinmanhan record is somewhat better but remains inadequate. Among non-mesonychians, Irdinmanhan hyaenodonts include two species of *Propterodon*, *P. morrisi* and *P. tongi*, the sinopanine *Sinopa jilinia*, and the limnocyonine *Prolaena parva* (*Matthew & Granger, 1924*; *Matthew & Granger, 1925*; *Xu et al., 1979*; *Tong & Lei, 1986*; *Lavrov, 1996*; *Liu & Huang, 2002*; *Morlo et al., 2014*). In addition to the machaeroidine *Apataelurus pishigouensis*, the last recorded oxyaenine, *Sarkastodon hetangensis*, occurs in the Irdinmanhan (*Tong & Lei, 1986*). Finally, Irdinmanhan miacoids are represented by three species, all questionably referred to *Miacis*: *M. boqinghensis*, *M. invictus*, and *M. lushiensis* (*Matthew & Granger, 1925*; *Chow, 1975*; *Tong & Lei, 1986*; *Qi, Zong & Wang, 1991*; *Huang, Tong & Wang, 1999*). Of these, only *Propterodon morrisi* and *Miacis lushiensis* are represented by multiple specimens (this may be in error for *M. lushiensis* as the size and morphology of referred material suggests the presence of multiple species).

Considering the limited nature of the Asian record, the presence of four genera shared between Uintan and Irdinmanhan faunas (*Harpagolestes*, *Apataelurus*, *Sinopa*, *Propterodon*) constitutes clear evidence for substantial exchange of carnivorous mammals during this interval. As noted above, *Prolaena* can be potentially added to this list although *Morlo & Gunnell (2003)* were skeptical of a relationship between Asian *Prolaena* and North American *Oxyaenodon*. Despite the assignment of species on both continents to a wastebasket "*Miacis*", there is less obvious overlap between miacoids, although "*Miacis*" *lushiensis* has been compared with Bridgerian "*M*". *hargeri* (*Tong & Lei, 1986*). Further study will be required to confirm this possibility and assess the potential for North American connections for other Irdinmanhan "*Miacis*". For the present, it is clear that investigations into the decline in North American hyaenodont diversity and coincident rise in carnivoraform diversity must consider the role of immigration in shaping the North American carnivore guild during the Uintan.

## CONCLUSIONS

The new species described in this work, *Propterodon witteri*, is the first known North American representative of the genus *Propterodon*. Comparisons of the new species with

other early and middle Eocene hypercarnivorous hyaenodonts support a link to Asian *Propterodon* and Hyaenodontinae more generally, a conclusion supported by the results of the phylogenetic analysis. The broader relationships of Hyaenodontinae are not well-resolved. Despite being supported by several phylogenetic assessments, a link to European *Oxyaenoides* is unlikely. An Asian origin for Hyaenodontinae is more likely, but better material of poorly known Linchan and Arshantan hyaenodonts is needed to test this hypothesis. Recognition of a Uintan hyaenodontine and an Irdinmanhan machaeroidine increases the evidence for dispersal of carnivorous mammals between Asia and North America during the late middle Eocene. Much of the apparent shift in North American carnivorous guilds, from "creodont" to carnivoramorphan dominated, may ultimately reflect the effects of this immigration rather than intrinsic processes within North American faunas.

**Institutional abbreviations**

| | |
|---|---|
| **AMNH FM** | Fossil Mammal Collection, American Museum of Natural History, New York, New York, USA |
| **CM** | Carnegie Museum of Natural History, Pittsburgh, Pennsylvania, USA |
| **HGL** | Hammada Gour Lazib, Algeria |
| **IVPP** | Institute of Vertebrate Paleontology and Paleoanthropology, Chinese Academy of Sciences, Beijing, China |
| **MCZ VPM** | Museum of Comparative Zoology, Harvard University, Cambridge, Massachusetts, USA |
| **MNHN.F.ERH** | Muséum National d'Histoire Naturelle, Rhône Basin Collection, Paris, France |
| **MNHN.F.RI** | Muséum National d'Histoire Naturelle, Rians Collection, Paris, France |
| **ZIN** | Zoological Institute, Russian Academy of Sciences, Saint Petersburg, Russia |

# ACKNOWLEDGEMENTS

Thanks to Jessica Cundiff (MCZ) for arranging the loan of MCZ VPM 19874. Initial photographs of the specimen were taken by S. Tomiya (Kyoto University). M. Borths (Duke University) graciously shared photographs of Asian species of *Propterodon*. Constructive reviews by M. Borths and A. Friscia greatly improved this manuscript.

## Funding

The author received no funding for this work.

## Competing Interests

The author declares there are no competing interests.

## Author Contributions

- Shawn P. Zack conceived and designed the experiments, performed the experiments, analyzed the data, contributed reagents/materials/analysis tools, prepared figures and/or tables, authored or reviewed drafts of the paper, approved the final draft.

## Data Availability

The data is available at MorphoBank, project id number P3489, http://morphobank.org/permalink/?P3489.

MCZ VPM 19874 is accessioned under this catalog number into the collections of the Museum of Comparative Zoology, Harvard University, Cambridge, Massachusetts, USA.

## New Species Registration

The following information was supplied regarding the registration of a newly described species:

Publication LSID: urn:lsid:zoobank.org:pub:CDA777EE-C052-4922-90DD-AAFD41D3F345

Propterodon witteri sp. nov. LSID: urn:lsid:zoobank.org:act:4D88F815-E7BE-4997-890F-59BC65A06A28

## Supplemental Information

Supplemental information for this article can be found online at http://dx.doi.org/10.7717/peerj.8136#supplemental-information.

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
