# Peer review of "The first North American Propterodon (Hyaenodonta: Hyaenodontidae), a new species from the late Uintan of Utah"

_PeerJ, doi:10.7717/peerj.8136_

## Round 0.1 · original submission · Minor Revisions

Dear Dr. Zack,

Thank you very much for this submission. I’m pleased to report that we have obtained two very helpful reviews from colleagues who are well-known experts on hyaenodonts. I agree with both reviewers that the paper is well done and relevant, and that it requires minor (to perhaps the lighter side of “major”) revisions. The reviewers complimented you on several aspects of your work, including your deep general knowledge and command of critical details, the usefulness to other workers of your extensive literature review, and your careful taxonomic analysis and comparisons of the material.

The reviewers also mentioned quite a few items that they thought would improve the paper. I found all their comments valuable, and for some you will need to weigh the positive impact versus the time needed to follow through on each suggestion. In general, you should not engage in extensive, time-consuming new analyses and figure drafting at this stage, but you should take every opportunity to improve the paper where possible and reasonable. Please consider and respond to each suggestion in your response letter, including my own at bottom. Here is my guidance-

Reviewer 1 made a list of minor corrections that will be straightforward to implement or respond to.

Reviewer 2 suggested some more substantive changes, as well as more minor but also important items. Among the substantive changes, at least the following should be addressed in your revision or response letter:

-The new combination Apataelurus pishigouensis definitely should go in the Systematic Paleontology section as suggested, with all basionyms, synonyms, and authorities included among all the standard formalities.
-Eliminate all “anagenetic” language (“derived from…”) about untestable ancestor-descendant relationships, as described by reviewer.
-More broad context about the Asia-America exchange as suggested (e.g., discussing more mammal groups that were involved and the timing of migrations) would help the paper find a larger audience. The reviewer also suggested adding a figure to illustrate the exchanges, which certainly would be nice but not required at this stage of your work.
-Locality map: I agree that the map is underpowered at the size given. On the other hand, I wouldn’t want you to enter a time-consuming effort to generate a high-end geologic map. Something in between seems appropriate, such as the addition of some geological information at perhaps smaller figure size.

-Improving TNT explanations to ensure reproducibility.
-Eliminate apparently contradictory text, for example that regarding the value of dental morphology characters in analyses as described by the reviewer.
-Consistent discussion and usage of OTUs and specific taxa and why included or not included in analyses.
-Eliminate “assertions” about the informative value of characters as described by the reviewer.

Additional comments:
It seems non-standard to include names of other taxa in a species diagnosis, which should be a stand-alone list of features that best distinguish a new taxon (without needing to name other taxa). If there is a precedent for this, please explain in your reply letter, otherwise modify the diagnosis to a standard format. "Largest known species" also does not seem to be a valid diagnostic character without measurements or other data that would remain useful if, say, an even larger species were found in the future.

Although your writing is strong, please be aware that PeerJ does not copy-edit papers (this helps keep the charges to authors reasonable). Please spend extra time combing your manuscript for any small grammatical errors and the like. To assist you, I attach an annotated pdf with a few items marked.

Kind regards,
Peter Wilf

·

Basic reporting

This paper is a valuable contribution to taxonomic, geographic, and temporal areas that needs more work. The author does well in his review of the literature and placing his work within the broader context of creodont evolution (a notoriously messy area of mammalian paleontology...). The author has an extensive background in the area, and he displays it in the thoroughness of his work. Only a few small revisions are required to make the paper ready for publication.

Experimental design

The research is relevant and meaningful, as outlined above.

Validity of the findings

There certainly was lots of exchange between North America and Eurasia throughout the Eocene and this paper adds another link between the continents.

Additional comments

A few small changes should be made before publication:
Line 42 - Unclear why there are two orphaned dashes in this line...
Line 52 - I believe you mean _Oxyaenodon_ not _Oxyaenoides_. Also, although you refer to _Sinopa_, I'm not sure that's a generally accepted synonymy of _Proviverra longipes_ (at least I couldn't find a reference to it). I'm not sure if a reference is needed here, or if you should just check on that.
Line 97 - add 'taxonomic' to read 'To test the taxonomic affinities...
Line 101 (and other places, including References) - The Borths & Stevens paper on _Simbakubwa_ is out, so replace with 2019.
Line 101 - missing a period after Rana et al.
Line 113 - remove 'was placed' and 'ensure'
Lines 125-126 - What do you mean by 'accompanying models' here? Data sets?
Line 127 - change 'base' to 'be based'
Line 129 - change 'used' to 'defined'
Line 141 - What do you mean by 'successional status' here? Ownership? Taxonomy?
Line 195 - Change 'for a series of' to 'based on'
Line 262 - change to read, '...cingulid that extends distally, even with the carnassial notch...'
Line 454 - change first 'of' to 'in'
References - Hooker 1989 is not cited in the paper.
Figure 1 - Seems like a lot of wasted space in this figure. Could you put an inset showing the position of Leota Quarry within the Uinta Basin?
Figure 3 - Maybe at the range of _Apataleurus kayi_ here since you talk about the genus.
Figure 4 - Could the inset of the occlusal view of the teeth be enlarged (with an appropriate scale bar)?
Figure 6 - Can you add labels indicating the clades you mention in the text: Limnocyoninae, Hyaenodontinae, Proviverrinae, Indohyaenodontinae, Hyainailourinae, Apterodontinae, Teratodontinae?

·

Basic reporting

Overall, this study represents an important contribution to the understanding of carnivore evolution in North American and Asia. Specifically, by identifying a new species of the otherwise Asian genus Propterodon it is evidence of an important exchange taking place between the continents. This new species is a significant data point for interpreting the diversity and ecology of carnivores during the Eocene in North America. This manuscript is also a testament to the importance of natural history collections as repositories and sources of new information on past and present ecosystems as the holotype was collected nearly eighty years ago.

This study is also exceedingly important because it reviews the history and taxonomy of Propterodon, a taxon that has been described in multiple languages over a century of work. This study will be a crucial resource for researchers investigating Eocene carnivores who need assistance hacking through the issues surrounding this important genus. This study supports the hypothesis that a species of Propterodon is sister-taxa of Hyaenodon - a speciose and wide-ranging hypercarnivorous genus. Propterodon is significant for interpreting the evolution of Hyaenodon, and this the first studies to directly address this important genus with a cladistic analysis.

The morphological descriptions are active, direct, and concise and the comparisons are thorough and appropriate, making a bulk of this paper acceptable as it is written. I only have a few concerns that I hope the author incorporates into a revision of the study to help make it more logically consistent and to ultimately make this an even more significant contribution. I also suggest a few figure modifications and additions that would illustrate the discussion featured in the manuscript. Finally, there are a few taxa mentioned in the discussion that are not included in the phylogenetic analysis. If possible, all hyaenodonts mentioned in the text in an evolutionary or biogeographic context should also be incorporated into the tree, or the author should explicitly describe why these taxa (Oxyaenodon in particular) were not incorporated into the phylogenetic analysis in the methods section.

Experimental design

The methods are appropriate to the question and the morphological descriptions are beautifully written. The comparisons are appropriate, barring a few suggestions in the General Comments section. I do think the methods used in TNT need to be made more specific so readers attempting to replicate the analysis can be confident they have followed the settings of the author when reproducing the study.

The major concern I have is that the methods section includes some discussion that undermines the larger conclusions of the manuscript. There is discussion in the methods about the superiority of this matrix over previously published matrices because it includes fewer correlated dental characters, but the criteria the author is using to determine what is a correlated character and what is an independent character is not explained.

Further, the author cites studies that argue dental morphology is unreliable as a source of phylogenetic information, then proceeds to name a new species based entirely on dental and proximal gnathic material, refer a species to a different genus based on dental morphology, and speculate on biogeographic exchange between two continents based on dental specimens. The discussion in the methods section caused me as the reader to wonder why the author trusted any of the succeeding comparative work if the author is so skeptical of the phylogenetic potential of dental morphology. The author's perspective on what qualifies as sufficient information to draw more than basic comparative conclusions should be expanded to make the following work logically consistent.

The discussion of OTUs should also be restructured to address the paraphyly of “Koholiinae” (a note expanded in General comments to the author). The methods section should also mention why relevant taxa evoked later in the manuscript are not included as OTUs, particularly Oxyaenodon.

Throughout the manuscript there are assertions about characters that are likely to be convergent and characters that are phylogenetically informative, but there is no method presented for how the author is making these determinations. It seems the author can either choose to make their assumptions more explicit, or the author can simply present the character-taxon matrix as they have chosen to construct it without making assertions about removing non-independent characters, a task that would seem difficult without a robust developmental perspective on genotypic correlation and its phenotypic expression.

Validity of the findings

The core of the study - the identification of an Asian genus in North American rocks – is important and valid. The larger context of exchange during the interval is also robust. One change I suggest is that the author spend less space identifying features that differentiate Oxyaenoides – a known and undisputed genus – from Propterodon – another uncontroversial genus and instead devote more space to contextualizing faunal exchange between North America and Asia during the late middle Eocene. The author focuses on the carnivore record, but a brief synopsis of other mammalian groups participating in this exchange would strengthen the utility of the manuscript for authors interested in larger-scale questions of ecological change.

Additional comments

Below are specific in-line comments for the author. They are a mix of practical and philosophical questions that will hopefully further strengthen the study:

Line 28: “…and appears unlikely.” This implies a likelihood or probabilistic method of phylogenetic inference was used. These methods were not employed in this study. In the methods section, while comparing the results of this study to many previous studies, the author should make it more clear that a different underlying model for phylogenetic inference is being used than has been employed by recent authors evaluating hyaenodont phylogeny. This language makes it sound like the author used Bayesian or likelihood methods, when they employed parsimony analysis.

Line 140: Borths and Stevens 2017 added deciduous dental characters to the character-taxon matrix. It is possible to test the phylogenetic position of Eoproviverra with a dp4 given published characters, or characters that can be modified. The selection of OTUs here is odd because it also doesn’t include all taxa relevant later in the discussion. Why is Oxyaenodon excluded from the analysis when it becomes relevant in the later discussion?

Line 147: The clade Koholiinae has not be recovered in recent analyses by Borths et al. Tinerhodon is resolved outside Hyaenodonta, Boualitomus and Lahimia are sister taxa, and Koholia is a very fragmentary specimen, though it is resolved along the stem of Hyainailouroidea. The blanket exclusion of discussion of this as a clade is not consistent with recent results. The absence of p1 is not known in Koholia because it is a fragmentary maxilla and the paracone of M1 may not be more distinctly lingual relative to the metacone than it is in Tritemnodon when wear is accounted for. The reasoning in this section should be reframed to note the possibility that Koholiinae is not monophyletic.

Line 167: Is there a reason the author refers to these as “minimum length trees” rather than “most parsimonious trees”? The latter is a more common term for referring to the resultant trees. Does this section imply there was a cap of 100 MPTs imposed in this analysis, or that the same number of MPTs were recovered by 100 separate analyses of the dataset? This description of the settings used in TNT should be more explicit. Based on this description, I am not confident I replicated the results via the same steps used by the author.

Line 222: The locality map is very simple and underutilizes space the color options available as a PeerJ publication. A basic geological map of Utah, or better a map of the basin with the Quarry situated in the larger sequence would be very useful. At present, the figure communicates little but political boundaries when it could communicate geological context.

Line 303: The author uses the phrase “closed trigonids” multiple times through the manuscript. At the first incidence of this morphology it would be useful to offer a brief explanation, describing how this is not synonymous with the presence of a large metaconid.

Line 323: Delete “below” or "before the end of"

Line 332: This is a confusing sentence that should be rephrased for clarity. Perhaps, “Unlike in Pyrocyon or Tritemnodon, m3 is distinctly larger than m2 in Furodon, Propterodon, and Oxyaenoides, a feature shared with P. witteri.”

Line 346: The structure of this sentence and the inclusion of the parentheticals makes it difficult to understand this section. I suggest the author restructure the sentence to make it easier to navigate the parentheses. For example, “Overall, Propterodon witteri displays a mixture of more derived morphology (open trigonids, trenchant talonids) and less derived morphologies (retained metaconids, elongate talonids) in comparison to Oxyaenoides.

Line 410: Note that Borths, Holroyd, and Seiffert 2016 and Borths and Seiffert 2017 used parsimony methods. The author’s comment holds in all of these cases with similar clades appearing, but low bootstrap support across much of the tree. It might support the point to mention these other parsimony results along with Rana et al. (2015).

Line 436: This discussion of the possible origins of Boritia is out of place in the results section as it speculates on relationships (Boritia + Prototomus) that are not resolved in the phylogenetic analysis. The results section should report what was resolved and move speculation on OTU origins to the discussion section.

Line 443: This section should be retitled “Hyaenodontine Sister-Taxa” or “Oxyaenoides and Hyaenodontinae” as it is essentially a description of the author’s skepticism that Oxyaenoides is a sister-taxon of Hyaenodontinae. This section amounts to an assertion that Oxyaenoides and Propterodon are distinct genera and Oxyaenoides is not the ancestor of Propterodon. I was not under the impression that any author explicitly argued that Oxyaenoides is the ancestor of Propterodon. No phylogenetic analysis can test for ancestor-descendant relationships. Humans are chimpanzees’ sister-taxon not their ancestor. Much of this section involves implicit and explicit discussion of ancestor-descendent relationships that cannot be evaluated given the patchy fossil record through this interval. This section can be used to express the author's skepticism that Oxyaenoides is the sister-taxon of Hyaenodontine, but the author should also note that in the phylogenetic analysis, the clade that includes Oxyaenoides and Matthodon is only one step removed from a sister-clade relationship to Hyaenodontinae.

Line 501: While the current organization makes sense, I wonder if it would be even easier for a reader to reference the new referral of Apataelurus pishigouensis if this was part of the Systematic Paleontology section. As formatted now, this important demonstration of the first Asian machaeroidine gets lost in the Discussion section. Another exceedingly useful addition would be a comparative figure illustrating Apataelurus pishigouensis and Apataelurus kayi. In addition to aiding the reader through the comparative section, such a figure would also help skimming readers recognize this paper includes the first Asian machaeroidine.

Line 502: Add a colon after Propterodon (“two additional Asian species of Propterodon:…”)

Line 565: “Among hyaenodonts, Oyaenodon dysodus does not appear to be derivable from Limnocyon potens” is a loaded statement with implied assumptions about the recovery of direct ancestor-descendent relationships. This early 20th century discussion contrasts with the author’s use of phylogenetic methods. It is possible to test whether the available morphological evidence supports a sister-taxon relationship between Oxyaenodon and Limnocyon, but the author does not use these species-level OTUs. As in the evaluation of the character differences between Oxyaenoides and Propterodon, the reader is left to discern the author's assumptions about morphological evolution. While important comparative commentary, such commentary does not yield reproducible results that can be built upon or tested by later researchers. One advantage of using cladisistic methods is researchers can be explicit in their models of evolution and can summarize comparative notes through a character-taxon matrix.

Figures: A figure like Figure 3 should be created to support the discussion of U1 and U2 faunas. The relevant taxa could also be incorporated into a more graphical presentation that uses a map, but a stratigraphic column that includes carnivoramorphans and other dispersing taxa would be very useful for the reader. Further, a figure, or at least an in-text, brief discussion of non carnivore exchanges during this interval would be exceedingly useful.

---

## Round 0.2 · Minor Revisions

Thank you for your patience and for this very careful and thoughtful revision, which addressed all concerns raised in the 1st review. General readers will appreciate the enhanced biogeographic discussion that you added. We have been fortunate to receive a second review from the original second reviewer, who recommended accepting the paper while noting some very minor edits that are needed. In addition to those changes, please respond to the following minor items:

1. The measurements in the diagnoses seem overly precise (tenth of a millimeter). If a specimen were found with dimensions 0.15 mm larger or smaller, it would be excluded from these taxa. Is that your intention? Please modify or justify.

2. Figure 1:
Please include another vertical bar with the standard names of epoch divisions and absolute age scale indicated. Most non-specialist readers are not automatically familiar with the ages of Eocene polarity chron boundaries or NALMA zonations.

The line color used for roads through Vernal is too light, please darken for visibility.

3. Figure 1 caption:
Please indicate the epochs of the formations listed to orient the reader (i.e., "middle Eocene Green River Fm.". etc.).

"geomagnetic chrons" -> "geomagnetic polarity chrons"

In subcaption A it would be best to cite the sources of the biozonation and pmag data more directly, i.e., "biochron boundaries (Prothero, 1996) and geomagnetic chrons. (Murphey et al., 2018)" (if that is correct). Also indicate whose measured section is shown.

4. Please proofread once more to screen for any typos or grammatical issues that could be present.

I look forward to seeing your revision very soon so that we can move this excellent work toward publication.

·

Basic reporting

The revised manuscript addresses many of the questions I had about the initial submitted draft. The revised manuscript preserves the clear anatomical description and the comparative considerations raised by these specimens.

The restructured manuscript flows more easily and I believe it will be easier for readers to find the information they need within the text. The revised figure is effective and overall I think this will be a very useful document for carnivore researchers in the future.

Experimental design

The details of the TnT analysis were restructured in a way that allowed me to confidently replicate the phylogenetic tree. The explanations for OTUs included and excluded from the analysis were sufficient for me to follow the author's process.

Validity of the findings

The discussion and importance of the discovery, particularly the emphasis on the exchange between Asia and North American during this interval, and the evidence that museum collections remain a rich source of biodiversity discovery, make this an important and novel study. I am looking forward to engaging with the author's discussion and conclusions in my own work.

Additional comments

I greatly appreciate the author's time in addressing the questions I raised in my initial review of the study. This is a very exciting discovery and I look forward to seeing it in print in PeerJ.

Two very minor typos:

Line 100: Replace “character” with “characters”

Line 251/253: Insert space between “tooth” and “row”? This may be a style issue, but this seems like an unusual compound word.

---

## Round 0.3 · accepted · Accept

Thank you for your excellent contribution to PeerJ.